# Intensive Urbanization, Urban Meteorology and Air Pollutants: Effects on the Temperature of a City in a Basin Geography

**DOI:** 10.3390/ijerph20053941

**Published:** 2023-02-22

**Authors:** Patricio Pacheco, Eduardo Mera, Voltaire Fuentes

**Affiliations:** Departamento de Física, Facultad de Ciencias Naturales, Matemáticas y Medio Ambiente, Universidad Tecnológica Metropolitana, Las Palmeras 3360, Ñuñoa, Santiago 7750000, Chile

**Keywords:** thermal conduction, discretization, time series, chaos, entropy, diseases

## Abstract

A qualitative study of thermal transfers is carried out from a record of measurements (time series) of meteorological variables (temperature, relative humidity and magnitude of wind speeds) and pollutants (PM_10_, PM_2.5_ and CO) in six localities located at different heights in the geographic basin of Santiago de Chile. The measurements were made in two periods, 2010–2013 and 2017–2020 (a total of 2,049,336 data), the last period coinciding with a process of intense urbanization, especially high-rise construction. The measurements, in the form of hourly time series, are analyzed on the one hand according to the theory of thermal conduction discretizing the differential equation of the temporal variation in the temperature and, on the other hand, through the theory of chaos that provides the entropies (S). Both procedures demonstrate, comparatively, that the last period of intense urbanization presents an increase in thermal transfers and temperature, which affects urban meteorology and makes it more complex. As shown by the chaotic analysis, there is a faster loss of information for the period 2017–2020. The consequences of the increase in temperature on human health and learning processes are studied.

## 1. Introduction

### 1.1. Thermal Islands

Heat islands are a phenomenon that occurs in those urban areas that experience higher temperatures than the surrounding areas due to human activity. The main cause is the accumulation of structures, such as buildings, sidewalks or asphalt, which absorb more heat and release it more slowly, unlike natural places such as forests, rivers or lakes. Added to this, heat and pollution are generated by traffic and industry. All this aggravates the consequences of climate change in cities and decreases the quality of life of its inhabitants [1,2]. In the last 20 years, associated with climate change in Chile, there have been heat waves, which have intensified and increased in duration. Heat waves are becoming more frequent, especially in the central zone of the country, which is where the capital of Chile, Santiago de Chile, is located in a basin geography. According to the Report on the Evolution of the Climate in Chile 2021, carried out by the Chilean Meteorological Directorate, the last 11 years have been the warmest on record. This shows the relevance of having relatively simple techniques that allow knowing the initial thermal condition of various communes before the arrival of heat waves. This condition is built by urban micrometeorology, pollutants and a geographical environment subjected to intensive urban densification.

The heat island effect is characterized by causing higher temperatures in cities than in their surroundings—towns, residential areas, agricultural areas, etc.—and is more pronounced at night, especially in winter. The average annual air temperature in a city with a million inhabitants or more can be between 1 °C and 3 °C higher during the day compared to its periphery, while at night this difference can even rise up to 10 °C [3,4,5].

### 1.2. Heat and Human Health

Urban heat islands and their contribution to the increase in temperature have negative consequences on people’s lives and the environment: increased energy consumption, impact on health [6,7,8], increased air pollution [9], impact on the economy [10].

Various initiatives have emerged to minimize the impacts of urban heat islands. Some of them are committed to sustainable urban development, as claimed by the United Nations Organization itself through the New Urban Agenda [11]. Some countries are promoting the investigation of old materials (revaluation of wood) and new ones for construction [12,13]. These have the potential to replace petroleum-derived plastics, in line with the concept of a zero-waste policy.

From the population’s perspective, high temperatures increase ozone levels and other air pollutants that aggravate cardiovascular and respiratory diseases. Likewise, levels of pollen and other aeroallergens are higher at high temperatures, which can trigger asthma episodes [7,8].

One of the changes in people that is related to heat is the increase in violent behavior. A mainstream psychology study [14] concludes that high temperatures enhance aggression by directly increasing feelings of hostility in people, and indirectly increasing violent thoughts. According to this research, heat increases aggressiveness because it induces a bad mood in people. This negative emotionality is accompanied by somatic consequences such as clenching of the fist, tachycardia, stomach discomfort, perspiration, etc. In the same way, if a person feeling hot is insulted, it is highly likely that they will respond with a greater insult. A hot and crowded dining room can lead to an exchange of insults, hitting and even more aggressive actions [15].

Cooperation, used daily by people to work together, not only depends on the willingness of people, but also on the temperature in the environment [15,16,17,18,19]. However, why does the heat affect work and productivity? Belkin [20] points out how the performance of people and their cognitive capacity (the mental processes that allow us to perform a task) also depend on the temperature indicated by the thermometer [20,21]. Belkin arranged a number of people separated into three different groups, each subjected to different temperatures. One group was subjected to a comfortable temperature, another to a neutral one, and another to a warm one. Each group was given the task of looking at a screen on which arrows appeared in different directions and indicating where the arrows were pointing. It was an exceptionally simple task that showed that the group that was in a comfortable temperature place could perform better than the group that was in a warm environment. In Chile, school dropout, school and urban violence have experienced a notable upturn in the last ten years. The Chilean Ministry of Education has implemented the strategy of providing educational resources to support educational communities immersed in school violence, direct intervention actions to assist educational establishments with critical cases and the creation of an advisory council. In its intentions, it declares that it is essential to understand the problem of coexistence, mental health and well-being in the school community in an educational way. These guidelines do not point directly to the effects of climate change and the increase in temperature on learning, since there are practically no studies in this regard.

## 2. Theoretical Bases

The exposed precedents are present, almost without exception, in all the cities of the world. This work is a comparative study from hourly time series of measurements, meteorological variables (temperature (T), relative humidity (RH) and magnitude of wind speed (WS)) and pollutants (PM_10_, PM_2.5_, CO), registered in six different locations, located at different heights and in two periods, 2010–2013 and 2017–2020, where the most recent period corresponds to intensive urban densification. All the locations are immersed in the same specific geographic morphology (Santiago de Chile). Two different time series analysis techniques will be applied: the theory of thermal conduction and the theory of chaos. The objective is to answer the question: is it possible to obtain indicators derived from both techniques, applied to time series, that analyze thermal transfer and that their predictions are similar, showing an increase in temperature and heat? With this purpose, the fundamentals of both techniques will be briefly developed.

### 2.1. Thermal Conduction

Heat conduction or energy transfer in the form of heat by conduction is a heat transfer process based on direct contact between bodies, without the exchange of matter: heat flows from a body with a higher temperature to another at a lower temperature in contact with the first.

#### 2.1.1. Physical and Mathematical Principles of the Heat Transfer Equation

The general conservation equation in a control volume can be derived considering the Reynolds control volume (CV) equation [22], as follows:(1)$$\frac{dN}{dt}=\frac{\partial }{\partial t}\int \rho \eta dV+\int \eta \rho \vec{v}\cdot d\vec{A}$$

By the Divergence Theorem:(2)$$\int \eta \rho \vec{v}\cdot d\vec{A}=\int \nabla \cdot \eta \rho \vec{v}dV$$

N is defined as the volume integral of the intensive quantity (per unit volume) n, that is, N=∫ndV
(3)$$\frac{dN}{dt}=\frac{d}{dt}\int ndV=\frac{\partial }{\partial t}\int \rho \eta dV+\int \nabla \cdot \eta \rho \vec{v}dV$$
(4)$$\frac{dn}{dt}=\frac{\partial \rho \eta }{\partial t}+\nabla \cdot \eta \rho \vec{v}$$
where η is defined as the heat content per unit mass $c_{P}$ T with T the absolute temperature.
(5)$$\frac{dn}{dt}=\frac{\partial \rho c_{P}T}{\partial t}+\nabla \cdot c_{P}T\rho \vec{v}\to \frac{dn}{dt}=\frac{\partial \rho c_{P}T}{\partial t}+\vec{v}\cdot \nabla c_{P}T\rho$$

and, for the incompressible:



(6)
$$\frac{\partial T}{\partial t}+\vec{v}\cdot \nabla T=\frac{1}{c_{p}\rho }\frac{dn}{dt}$$



Natural or free convection is heat transported by the velocity of the fluid caused by instabilities in temperature, for example, a hot fluid below a cold fluid. Advection or forced convection is the heat transported by the velocity of the fluid that is caused by (i) shear stress gradients, (ii) elevation, (iii) pressure imposed on the flow field. Thus, the advection term $\vec{v}\cdot \nabla T$ indicates the transport process of an atmospheric property, such as temperature, due to the effect of the wind speed ($\vec{v}=u\hat{i}+v\hat{j}+w\hat{k}$).
(7)$$\frac{dn}{dt}=\frac{dq}{dt}+R^{\prime }+S_{\tau }^{\prime }+S_{C}^{\prime }$$
where dq/dt is heat conduction, R′ is thermal radiation, $S_{\tau }^{\prime }$ is heat generation by mechanical dissipation (viscous dissipation) and $S_{C}^{\prime }$ is heat exchange due to chemical reactions.

The conduction of heat can be established as follows:(8)$${\vec{N}}_{C}=-K\nabla T$$
(9)$$dq=\nabla q\cdot d\vec{r}\to \frac{dq}{dt}=\nabla q\cdot \frac{d\vec{r}}{dt}=\nabla q\cdot \vec{v}=\nabla \cdot q\vec{v}=\nabla \cdot {\vec{N}}_{C}$$
(10)$$\frac{dn}{dt}=\frac{dq}{dt}+R^{\prime }+S_{\tau }^{\prime }+S_{C}^{\prime }$$
the quantities R′, $S_{\tau }^{\prime }$ and $S_{C}^{\prime }$ are divided by the volume (in the International System m^3^). Writing:$\frac{dq}{dt}=\nabla \cdot {\vec{N}}_{C}=\nabla \cdot K\nabla T=K\nabla^{2}T$ (assuming that the addition of thermal energy is what increases the temperature), is obtained:(11)$$\frac{\partial T}{\partial t}+\vec{v}\cdot \nabla T=\frac{1}{c_{P}\rho }\left(K\nabla^{2}T{+R}^{\prime }+S_{\tau }^{\prime }+S_{C}^{\prime }\right)$$
(12)$$\frac{\partial T}{\partial t}+\vec{v}\cdot \nabla T=\left(\frac{K}{c_{P}\rho }\nabla^{2}T+\frac{S_{\tau }^{\prime }}{c_{P}\rho }+\frac{R^{\prime }}{c_{P}\rho }+\frac{S_{C}^{\prime }}{c_{P}\rho }\right)$$

The differential equation of heat [22]:(13)$$\frac{\partial T}{\partial t}+\overset{\begin{array}{c} \mathrm{The}\;\mathrm{term}\;\mathrm{represents}\\ \mathrm{heat}\;\mathrm{transport}\;\mathrm{due}\;\mathrm{to}\\ \mathrm{convection}\;\mathrm{or}\;\mathrm{advection}. \end{array}}{\overbrace{\vec{v}\cdot \nabla T}=}\left(\underset{\begin{array}{c} \mathrm{It}\;\mathrm{is}\;\mathrm{the}\;\mathrm{term}\\ \mathrm{of}\;\mathrm{conduction}\\ \mathrm{or}\;\mathrm{diffusion}\\ \mathrm{heat} \end{array}}{\underbrace{\frac{k}{\rho c_{p}}\nabla^{2}T}}+\underset{\begin{array}{c} \mathrm{heat}\\ \mathrm{generation}\\ \mathrm{through}\\ \mathrm{shear}\;\mathrm{stress} \end{array}}{\underbrace{\frac{\emptyset }{\rho c_{p}}}}+\underset{\begin{array}{c} \mathrm{term}\;\mathrm{of}\\ \mathrm{radiation} \end{array}}{\underbrace{R}}+\underset{\begin{array}{c} \mathrm{chemical}\\ \mathrm{reaction}\mathrm{term} \end{array}}{\underbrace{S_{c}}}\right)$$
with $\emptyset =-S_{\tau }^{\prime }$ = viscous dissipation term [23]:$$\emptyset =2\left({\left(\frac{\partial u}{\partial x}\right)}^{2}+{\left(\frac{\partial v}{\partial y}\right)}^{2}+{\left(\frac{\partial w}{\partial z}\right)}^{2}\right)+{\left(\frac{\partial u}{\partial y}+\frac{\partial v}{\partial x}\right)}^{2}+{\left(\frac{\partial u}{\partial z}+\frac{\partial w}{\partial x}\right)}^{2}+{\left(\frac{\partial u}{\partial z}+\frac{\partial w}{\partial y}\right)}^{2}$$
(14)$$\frac{\partial T}{\partial t}=\overset{\begin{array}{c} \mathrm{The}\;\mathrm{term}\;\mathrm{represents}\\ \mathrm{heat}\;\mathrm{transport}\;\mathrm{due}\;\mathrm{to}\\ \mathrm{convection}\;\mathrm{or}\;\mathrm{advection}. \end{array}}{-\overbrace{\vec{v}\cdot \nabla T}+}\left(\underset{\begin{array}{c} \mathrm{It}\;\mathrm{is}\;\mathrm{the}\;\mathrm{term}\\ \mathrm{of}\;\mathrm{conduction}\\ \mathrm{or}\;\mathrm{diffusion}\\ \mathrm{heat} \end{array}}{\underbrace{\frac{k}{\rho c_{p}}\nabla^{2}T}}+\underset{\begin{array}{c} \mathrm{heat}\\ \mathrm{generation}\\ \mathrm{through}\\ \mathrm{shear}\;\mathrm{stress} \end{array}}{\underbrace{\frac{\emptyset }{\rho c_{p}}}}+\underset{\begin{array}{c} \mathrm{term}\;\mathrm{of}\\ \mathrm{radiation} \end{array}}{\underbrace{R}}+\underset{\begin{array}{c} \mathrm{chemical}\\ \mathrm{reaction}\mathrm{term} \end{array}}{\underbrace{S_{c}}}\right)$$

It can be written as [13]:(15)$$\frac{\partial T}{\partial t}+u\frac{\partial T}{\partial x}+v\frac{\partial T}{\partial y}+w\frac{\partial T}{\partial z}=\alpha \left(\frac{\partial^{2}T}{{\partial x}^{2}}+\frac{\partial^{2}T}{{\partial y}^{2}}+\frac{\partial^{2}T}{{\partial z}^{2}}\right)+\frac{\emptyset }{\rho c_{p}}+R+S_{c}$$

The partial differential equation (PDE) of heat transfer is classified as parabolic (y = ax^2^ + bx + c). This type of equation allows solving the so-called propagation problems, which are transient problems where the solution of the partial differential equation is required over an open domain, subject to prescribed initial and boundary conditions. The most common examples of these problems include heat conduction problems, diffusion problems and, in general, problems where the solution changes with time.

#### 2.1.2. Discretization of the Heat Transfer Equation

Writing Equation (15) in the form:(16)$$\frac{\partial T}{\partial t}=\alpha \left(\frac{\partial^{2}T}{{\partial x}^{2}}+\frac{\partial^{2}T}{{\partial y}^{2}}+\frac{\partial^{2}T}{{\partial z}^{2}}\right)+(-1)(u\frac{\partial T}{\partial x}+v\frac{\partial T}{\partial y}+w\frac{\partial T}{\partial z})+\frac{\emptyset }{\rho c_{p}}+R+S_{c}$$

Equation (16) can be discretized to analyze its behavior, in an exploratory way, using numerical methods. First the numerical coefficients are reduced:

α = coefficient of thermal diffusivity = $\frac{k}{\rho \;c_{p}}$ where k is the conductivity of the substance (for the atmosphere it is 0.02 W/(m/K) at the room temperature ambient (20–25 °C)), ρ is the density of air =1.29 kg /m^3^, C_P,_ the specific heat at constant air pressure (25 °C), is 1012 J /kg K
$$\alpha =\frac{k}{\rho \;c_{p}}=\frac{0.02\frac{W}{mK}}{1.29\frac{kg}{m^{3}}1012\frac{J}{kgK}}=1.53*10^{-5}\frac{m^{2}}{s}$$
(17)$$\frac{\partial T}{\partial t}=\frac{T_{i}^{l+1}-T_{i}^{l}}{∆t}$$

If z-dependence (one-dimensional) is considered, in a very rough first approximation:(18)$$\frac{\partial T}{\partial z}=\frac{T_{i-1}^{l}-T_{i+1}^{l}}{2∆z}$$
and:(19)$$\frac{\partial^{2}T}{\partial z^{2}}=\frac{T_{i-1}^{l}-2T_{i}^{l}+T_{i+1}^{l}}{∆z^{2}}$$
replacing:(20)$$\frac{T_{i}^{l+1}-T_{i}^{l}}{∆t}=\alpha \frac{T_{i-1}^{l}-2T_{i}^{l}+T_{i+1}^{l}}{∆z^{2}}+\left(-1\right)w\frac{T_{i-1}^{l}-T_{i+1}^{l}}{2∆z}+\frac{\emptyset }{\rho c_{p}}+R+S_{c}$$
(21)$$\frac{T_{i}^{l+1}}{∆t}=\frac{T_{i}^{l}}{∆t}+\alpha \frac{T_{i-1}^{l}-2T_{i}^{l}+T_{i+1}^{l}}{∆z^{2}}+\left(-1\right)w\frac{T_{i-1}^{l}-T_{i+1}^{l}}{2∆z}+\frac{\emptyset }{\rho c_{p}}+R+S_{c}$$
(22)$$T_{i}^{l+1}=T_{i}^{l}+\alpha \frac{T_{i-1}^{l}-2T_{i}^{l}+T_{i+1}^{l}}{∆z^{2}}∆t+\left(-1\right)w\frac{T_{i-1}^{l}-T_{i+1}^{l}}{2∆z}∆t+\left(\frac{\emptyset }{\rho c_{p}}+R+S_{c}\right)∆t$$

The equation that can be written is:(23)$$\gamma ∆t=\left(\frac{\emptyset }{\rho c_{p}}+R+S_{c}\right)∆t=T_{i}^{l+1}-{\;T}_{i}^{l}-\alpha \frac{T_{i-1}^{l}-2T_{i}^{l}+T_{i+1}^{l}}{∆z^{2}}∆t+w\frac{T_{i-1}^{l}-T_{i+1}^{l}}{2∆z}∆t$$

Solving γ from Equation (23), with ∆t = 1 h (for hourly time series with 28,643 data each), ∆z = 3 m (height in meters from the ground) and transforming the units of α and w, we obtain γ for each commune (C):(24)$$\gamma_{C}={(T}_{i}^{l+1}-{\;T}_{i}^{l})/(1)+(-1.53*10^{-5}\frac{T_{i-1}^{l}-2T_{i}^{l}+T_{i+1}^{l}}{9}+w\frac{T_{i-1}^{l}-T_{i+1}^{l}}{6})*3600\left[\frac{{}^{\circ}C}{h}\right]$$

Vertical velocities, w, constitute a significant part of the dynamics in the atmosphere. They maintain the air movements between adjacent levels in connection, providing direct communication between the existing flows at such levels. Thus, any realistic model of atmospheric motion, designed for both theoretical and forecasting purposes, must take into account the vertical component of motion. The vertical speed in large-scale movements is of the order of a few centimeters per second. From the continuity equation, it can be deduced that the horizontal divergence is of the order of 10^−4^ s^−1^. This value of the horizontal divergence is consistent with the value found in the vorticity equation. However, in small-scale motion (where the horizontal scale is of the order of 10 km), the vertical velocities turn out to be almost the same magnitude as the horizontal wind speed (5.3–7.4 m/s, moderate on the Beaufort scale) [24,25]. Figure 1 shows the hourly characteristic distribution of the magnitude of wind speed (V) in the study period of 3.25 years, the average value is 1.7177 m/s:

Figure 2 shows the characteristic temperature distribution for the 3.25-year period of the Puente Alto commune (average temperature in the study period 14.69 °C):

$\frac{\emptyset }{\rho c_{p}}\to 0$, represents the heat dissipation through shear stress, which is a small term compared to $R+S_{c}$.

$S_{c},$ provides the energy variation in the chemical reactions since all of them are accompanied by energy exchanges, either because they give off energy or because they absorb energy. Chemical reactions can be carried out at a constant volume or constant pressure. However, most of the reactions take place at a constant pressure as they occur in vessels open to the atmosphere (where the pressure is atmospheric). These processes when carried out at a constant pressure are called isobaric.

R = R’/Volumen represents the thermal radiation, R’, created or emitted by a point in the volume of a fluid, given that the temperature of the body is greater than zero. Two bodies that are radiating heat can exchange heat and no medium is required for the thermal energy to be exchanged by the radiating bodies (σ is the Stephan–Boltzmann constant which is 5.67 × 10 ^−8^ W /m^2^ k ^4^) ($R’=e\sigma AT^{4}$).

It is observed that, of the total solar radiation that reaches Earth from the sun, 30% is reflected (albedo) by clouds, the Earth’s surface and atmosphere (gases, dust...), 25% is absorbed by the atmosphere (due to the ozone layer (3%), water vapor and air particles (17% both) and clouds (5%)) and 45% is absorbed by the surface (75% from the oceans and 25% from the continents). The infrared radiation will leave the surface, slowly and gradually, toward the atmosphere in the form of latent heat associated with evaporation [26]. The radiation present in the atmosphere (either that absorbed by the atmosphere or that received from the Earth’s surface and that ends up returning to the atmosphere) is reintegrated into space in the form of long-wave radiation. Figure 3 shows the presence, in a volume element, of R, $S_{\tau }$ y $S_{C}$:

### 2.2. Kolmogorov Entropy

#### 2.2.1. Temporal Variation in Volumetric Entropy

Using the first law of thermodynamics, the continuity equation and the component balance equation and rewriting the local derivative of S = V·s, where V is the volume, the rate of entropy emission per unit volume is determined, s, from equation [22,27]:(25)$$\frac{\partial s}{\partial t}=-\nabla \cdot {\vec{J}}_{S}+\sigma$$
(26)$$\frac{\partial s}{\partial t}=-\nabla \cdot \left(s\vec{v}+\frac{1}{T}{\vec{J}}_{\mathrm{Heat}\;\mathrm{flux}}-\sum_{j=1}^{\;r}\frac{\mu_{j}}{T}{\vec{J}}_{,\mathrm{mass}\;\mathrm{diffusive}\;\mathrm{flow}\;j}\right)+\sigma$$
σ is called the term for the production of entropy toward the surroundings. This expression tells us that the temporal variation in the volumetric entropy associated with the substance depends on its speed, the heat fluxes associated with it, the diffusive mass flux and the entropy production. If the maximum ∂s/∂t is known for each time series (T, RH, WS, PM10, PM2.5, CO) in each location according to the two study periods 2010–2013 and 2017–2020, the variation in entropic transfer between periods and by locality can be estimated.

#### 2.2.2. Kolmogorov Entropy and Loss Information

According to Farmer, one of the essential differences between chaotic and predictable behavior is that chaotic trajectories continuously generate new information while predictable trajectories do not [28,29]. Metric entropy makes this notion more rigorous. In addition to providing a good definition of chaos, metric entropy provides a quantitative way to describe how chaotic a dynamical system is. In the Kolmogorov entropy [30,31], S_K_ is the average loss of information [32] when “l” (side of the cell in units of information) and τ (time) become infinitesimal [33]:(27)$$S_{K}=-\underset{\tau \to 0}{lim}\underset{l\to 0}{lim}\underset{n\to \infty }{lim}\frac{l}{n\tau }\sum_{0....n}P_{o....n}\log P_{0.....n}$$

S_K_ has units of information bits per second and bits per iteration in the case of a discrete system [34,35]. The limit process of Equation (1) follows the order: (i) n → ∞, (ii) l → 0, canceling the dependency of the selected partition (n is the number of cells or partitions) and (iii) τ → 0, for continuous systems. The Kolmogorov entropy difference, ΔS_K_ = S_Kn+1_ − S_Kn_, between two neighboring cells, gives the required complementary information about the cell (i_n+1_) in which the system will be in the future. The difference gives the loss of information, in time, of the system. In summary, for the calculation of the Kolmogorov entropy, it is first verified that the entropy is between zero and infinity (0 < S_K_ < ∞), which allows verifying the presence of chaotic behavior. If the Kolmogorov entropy is equal to 0, no information is lost, and the system is regular and predictable. If S_K_ = ∞, the system is completely random, and it is impossible to make any predictions. Secondly, the amount of information required to predict the future behavior of two interacting systems is determined, in this case, by the atmosphere (represented by meteorological variables) and the hourly concentration of pollutants. In this way, the rate at which the system loses (or outdates) information over time can be estimated. Finally, the horizon of maximum predictability of the system can be established. This horizon is a limit frontier from which it is not possible to make predictions or formulate new scenarios [31].

The loss of information can be determined according to the equation:(28)$$<∆I>=<I_{NEW}-I_{OLD}>=\frac{-\lambda \left(i_{0}\left(t\right)\right)}{log2}$$
λ is the Lyapunov exponent, <ΔI> in [bits/hr] is the loss information. Two types of <ΔI> can be calculated: one for the contribution of pollutants and another for the sum of the loss of information of each meteorological variable (MV): temperature (T), magnitude of wind speed (WS) and relative humidity (RH) [33].

#### 2.2.3. Flowchart

Figure 4 represents the process that allows verifying the chaoticity of each time series (the total is 72 for the two study periods, 2010–2013 and 2017–2020) [33]:

## 3. Materials and Methods

### 3.1. Area Where the Measurements Were Made

The capital city of Chile, Santiago de Chile, is located at 33.5° S and 70.8° W. It contains a population of 7,508,334 inhabitants, which represents 40% of the total population of the country, on a surface of approximately 641 km^2^. It is located in the middle of the country, at a height of about 520 m.a.s.l. The altitude above sea level increases from west to east. It is surrounded by two mountain chains: the Andes and the Chilean Coastal Range. Its climate is Mediterranean, Figure 5. The driest and warmest months are from December to February, reaching maximum temperatures of about 35 °C in the shade (air temperature in the sun), Figure 2. Given its topography and the dominant meteorological conditions, there is, in general, a strong horizontal and vertical dispersion of pollutants generated by an important number of sources in the city (heating, vehicles, industries, etc.), especially during fall (20 March–21 June) and winter (21 June–23 September) [36]. The emissions have a tendency to increase given the also increasing population density, which implies an increase in fixed and mobile sources. In addition, the number of vehicles has increased rapidly in recent years. The automotive fleet in Santiago de Chile has added about 750,000 vehicles since 2010, totaling today about 2.15 million cars, according to the latest data available, for 2018, from the National Institute of Statistics of Chile (INE, in Spanish). Growth will continue, and by 2025, another 546,000 units would be added, which implies a 25% increase in that period.

### 3.2. The Data

The measurements for the 2017–2020 period have the same amount as those for the 2010–2013 period and were collected from the MACAM III Network of the National Air Quality Information System, under the Ministry of the Environment of Chile [27,37]. The data correspond to 3.25 years or 39 months of measurements of PM10, PM2.5, CO, temperature (T), relative humidity (HR) and wind speed magnitude (WV), totaling 28,463 data points for each variable, giving a total of 1,024,668 measurements for all communes in the 2017–2020 period, Table 1. The temporal resolution of the time series is one hour.

## 4. Results

### 4.1. Discretization

Using Table 2 for the average magnitude of the wind speed for each time series and location for each study period:

Applying Equation (24) to the temperature time series for each location, with the elevation in the measuring instruments with respect to sea level (height), for the two periods of the study, and considering that ${\bar{\gamma }}_{C}=\bar{\frac{\emptyset }{\rho c_{p}}+R+S_{C}}=\mathrm{Average}\left(\frac{K}{h}\right)$, Table 3 and Table 4 are obtained.

Table 5 allows, through columns 2, 3 and 4, to visualize the behavior of the average value of ${\overset{-}{\gamma }}_{C}$ according to the height, period and commune of measurement, Figure 6.

Figure 6 represents ${\overset{-}{\gamma }}_{C}$ (K/h) versus h (m):

**Figure 6 ijerph-20-03941-f006:**
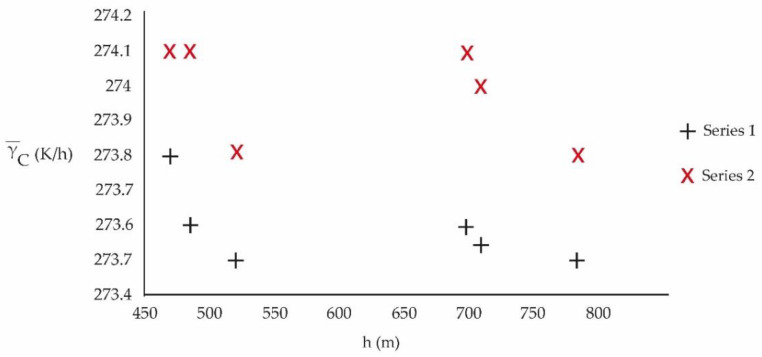
Temperature emission per hour, ${\overset{-}{\gamma }}_{C}$, according to height (masl), measurement location and for each study period (Series 1: 2010–2013 and Series 2: 2017–2020).

Figure 6 shows that in the period 2017–2020, of intensive urbanization with a lot of high-rise buildings, the hourly rate of temperature delivery, ${\overset{-}{\gamma }}_{C}$, to the atmosphere is higher than in the period 2010–2013.

### 4.2. Irreversible Processes

From the perspective of localized measurements carried out in a geographical basin (Table A1 and Table A2 of the Appendix A), and as a consequence of applying Clausius, the following is obtained:(29)$$\delta S=\frac{\delta Q}{\overset{-}{T}}\to \delta Q=\overset{-}{T}\delta S\to \;\frac{\delta S}{dt}=\frac{1}{\overset{-}{T}}\frac{\delta Q}{dt}$$

It is also possible to express the atmospheric pressure according to the formula:

$P=n_{V}K_{B}T=n_{oV}K_{B}Te^{-\frac{mgy}{K_{B}T}}$, k_B_ = 1.38 × 10^−23^ (J/K) = Boltzmann’s constant, but the heights of this study, with respect to sea level, indicate that P_0_ = $n_{oV}K_{B}T$_0_
~ 1.1013 × 10^5^ [N/m^2^] at the ambient temperature where the atmospheric density is approximately n_0_~2.69 × 1025 (molecules/m^3^). Through the Clausius relation, according to Equation (29), $\frac{\delta Q}{dt}=\overset{-}{T}\frac{\delta S}{dt}$ with $T_{K}=273.15+T_{{}^{\circ}C}$ is obtained.

The Kolmogorov entropy (δS/dt) of each time series of the meteorological variable, S_MV_, and of the pollutants, S_P_, in the six measurement locations for both periods were extracted from Table A1 and Table A2 of the Appendix A and appear in Table 5 and Table 6 with the height of each location and the average temperature of the registration period. Building the difference:(30)$$∆={\left(\frac{\delta Q}{dt}\right)}_{P}-{\left(\frac{\delta Q}{dt}\right)}_{MV}$$
and using Table A1 and Table A2 (from the Appendix A), the summary of the variables of interest appears in Table 6 and Table 7, which contains the average temperature for each measurement location of the study period.

When calculating the temperature difference in both periods, it shows an incremental trend ∆T=16.20 °C−15.64 °C=0.56 °C. Table 8 summarizes the height and Δ according to periods 2010–2013 and 2017–2020:

Figure 7 represents the difference Δ between the heat variation in time of the pollutants and the heat variation in time of the meteorological variables according to the different measurement locations:

Figure 7 shows that Δ is higher in the period 2017–2020 when compared to the period 2010–2013. This demonstrates a greater entropic flow in the period 2017–2020. Using the data from Table 6 of the reference [37], Table 9 is constructed, which provides the loss of information according to the periods 2010–2013 and 2017–2020:

Figure 8 allows us to compare the periods with the greatest and least loss of information:

## 5. Discussion

The localities considered in this study correspond to areas with great housing development, mainly high rise, favored by a policy of urban densification. The comparative study reveals that urban densification contributes to the increase in temperature. Although mitigation policies have been developed for a decade, they have not been effective and the problem has been growing. In addition, the main construction material used is high albedo concrete [25,27]. This is a sign that the changes that must be applied for the development of cities must be very profound and even consider geomorphological factors [33]. The variable ${\overset{-}{\gamma }}_{C}$ shows, in a very first approximation, the effect of heat dissipation by viscosity, by chemical reactions and by thermal radiation. Figure 6 shows the approximate average magnitude of the thermal transfer, in K/h, toward the environment of each one of the study periods. From the figure, it can be seen that the highest emission corresponds to the 2017–2020 period, with more intense urbanization, compared to the 2010–2013 period.

Figure 7 presents a high entropic transfer in the proximity to the ground, for the period 2017–2020, of intensive urbanization with a change in the roughness of the soils, which harmonizes with an increase in turbulence and is aligned with the cascade effect of Kolmogorov [38]. Between 450 m and 550 m, the rate of change in Δ with the height is positive for the period 2010–2013 and negative for the period 2017–2020. Figure 8 indicates that the loss of information for the period 2017–2020 is faster. This is compatible with a more turbulent and less predictable current regime compared to 2010–2013. Small changes in urban meteorology have a large effect on its future behavior. Based on Figure 7, which shows the difference in entropic transfer between the study periods, it is found that in the period 2017–2020, the urban densification system, by incorporating more pollutants, provides more entropy to urban meteorology.

Both methods, discretized thermal conduction and chaos, show that in the lowest altitude locations the variation in ${\overset{-}{\gamma }}_{C}$ and Δ is more dynamic.

From a sanitary point of view, the communes belonging to Santiago de Chile called Parque O’higgins (EMN), Puente Alto (EMS) and Pudahuel (EMO), which are the ones that are at the lowest altitude with respect to sea level, show the stronger variations in ${\overset{-}{\gamma }}_{C}$ and Δ. This group of three communes, with very high urban densification, contains the commune, EMS, which is the one with the highest urban densification in the country and had the highest number of SARS-CoV-2 infections [39,40].

Chile is experiencing a period of long drought and a weakening, due to climate change, in the seasons of the year (autumn and spring) that has been very marked. Santiago de Chile, located in a geographic basin, is not the exception to what is added by atmospheric pollution and urban densification [36,41,42]. Many of its peripheral communes (not considered in this study) have become exceptionally dry (Lampa, Til-Til, for example).

In [43], the results show that the total heat load of the xeric landscape (the vegetation and plant associations specifically adapted for life in a dry environment) is notably higher than the heat of the grassy landscape. For sectors with high urban densification, studies [44] indicate that it is necessary to improve the design of buildings and decision-making considering the urban microclimate. A mediator analysis indicates that environmental identity should be supported by forming connectivity networks between sustainable leadership and the effects of the environmental impact. This mechanism can not only advance the literature on sustainable development but should also help companies achieve sustainable development through the adoption of environmental innovation strategies [45]. Other research shows that the conceptualization and measurement of urban–rural resilience can be linked in the urban–rural system. These empirical findings reveal the impact of rapid urbanization on urban–rural resilience in recent years, which is common in many cities. Furthermore, the results of spatial heterogeneity could be used as a policy reference to develop specific resilience strategies in the study region [46]. The treatment of all types of waste produced by people should favor the development of more exhaustive control procedures for emissions, especially heavy metals into the atmosphere. Reducing electronic waste to its most elementary parts (considering that electro mobility is coming) would promote the global standardization of the dismantling of this modern garbage [47].

The results obtained confirm the generation of this thermal anomaly in the geographical morphology of the study: focused areas of heat emission, in the city, which cause an increase in temperature. This increase is largely caused by human activity and ends up harming the very people who created it. Recently, extreme events of a temperature increase due to heat waves (climate change) reached Europe, severely affected the health of many older adults [48,49] and was responsible for a large number of deaths. The effects on people’s health are very complex since they encompass their psychology, emotionality, biology and work, producing, in addition, a disturbance in the willingness to learn in students [50,51,52].

## 6. Conclusions

The methods applied to the study of the time series show, qualitatively, similar consequences. The discretization method is applied to the temperature time series and for this study an estimated value of the vertical component of the magnitude of the wind speed (WS) is used. WS is the average, according to locality, for the 3.25 years of measurements and the assumption that the magnitude of the vertical wind is of the order of the horizontal component of the velocity. The result indicates that the period 2017–2020 had an hourly temperature emission rate to the environment greater than the period 2010–2013, which is compatible with the comparison of the average temperatures measured between the two periods: ${\overset{-}{T}}_{2010-2013}=15.64\;{}^{\circ}C$ and ${\overset{-}{T}}_{2017-2020}=16.20\;{}^{\circ}C$, showing an increase of 0.56 °C. While the averages for the six study communes of ${\overset{-}{\gamma }}_{2010-2013}=273.59\;K/h$ and of ${\overset{-}{\gamma }}_{2017-2020}=274.00\;K/h$, respectively, with an increase of 0.41 °C/h.

The technique applied by the chaos theory is more complex from the point of view that it uses, in its analysis, the 72 time series that contain the data of the urban meteorology (magnitude of wind speed, relative humidity and temperature) and that of the air pollutants (PM10, PM2.5, CO) of the two periods considered in this work. The reason is that it is necessary to build the C_K_ indicator that uses the entropies of the six variables measured in each location. These entropies, for the urban meteorology and for the pollutants, make it possible to estimate entropic fluxes, showing that those of the 2017–2020 period are higher than those of the 2010–2013 period. That is, the chaotic technique shows the ability to quantify the effect of intensive urbanization and conclude that it is conducive to heat islands that act as thermal emitters, increasing the temperature in the atmosphere in the geographical basin. A relevant tangential aspect of chaos theory is that it requires minimal a priori assumptions.

The analyses of the empirical database used (a total of 2,049,336 data), measured in two periods of 3.25 years in locations at different heights in a specific geomorphology (basin) subjected to intensive urban densification in the most current period, confirm the urban densification as a significant cause of heat and temperature increase. According to the documents cited in this study, the harmful effect that this causes on human health in psychological, emotional and biological aspects has been documented, one of the most affected being the willingness of students to learn. A relevant aspect of all teaching is the modification of the behaviors that originated from the problems. This allows us to glimpse the magnitude of the challenge: if inadequate environmental conditions favor insufficient learning, it will be difficult to rethink cities in a way that is even hard to imagine.

## Figures and Tables

**Figure 1 ijerph-20-03941-f001:**
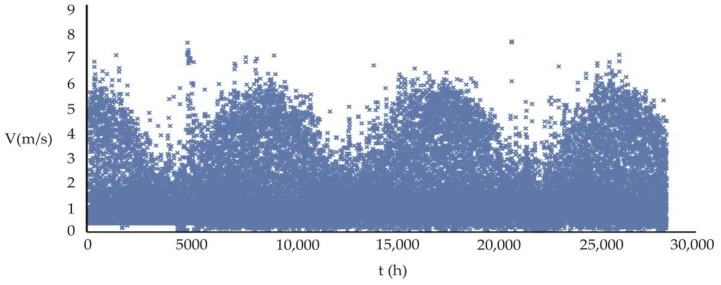
Distribution of the 28,463 hourly wind speed (WS) magnitudes for the period 2010–2013 for the Puente Alto commune. The Figure is similar for the other five communes in the study.

**Figure 2 ijerph-20-03941-f002:**
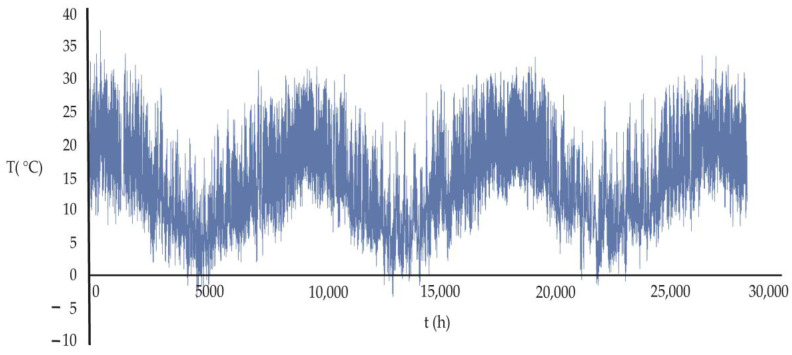
Distribution of the 28,463 hourly temperatures of the period 2010–2013 for the Puente Alto commune. The data distribution of the magnitude of the wind speed over time, for the other five communes, is similar to that shown in the figure.

**Figure 3 ijerph-20-03941-f003:**
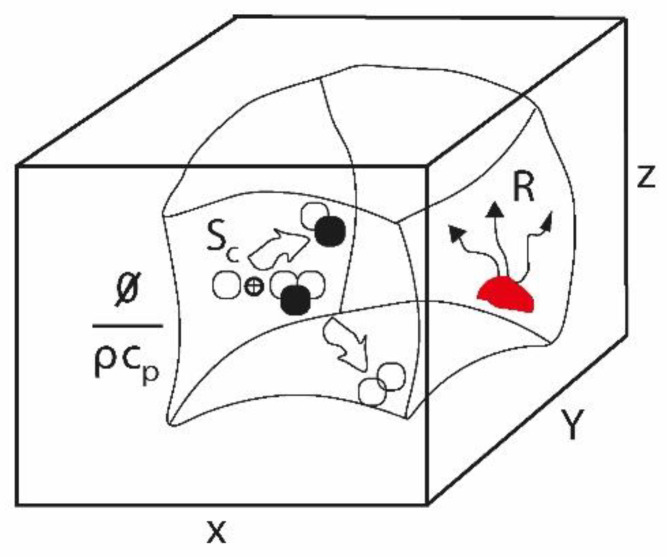
Three-dimensional representation in a volume element of heat dissipation through shear stress ($\frac{\emptyset }{\rho c_{p}}$), energy variation in chemical reactions ($S_{c}$ ) and thermal radiation (R).

**Figure 4 ijerph-20-03941-f004:**
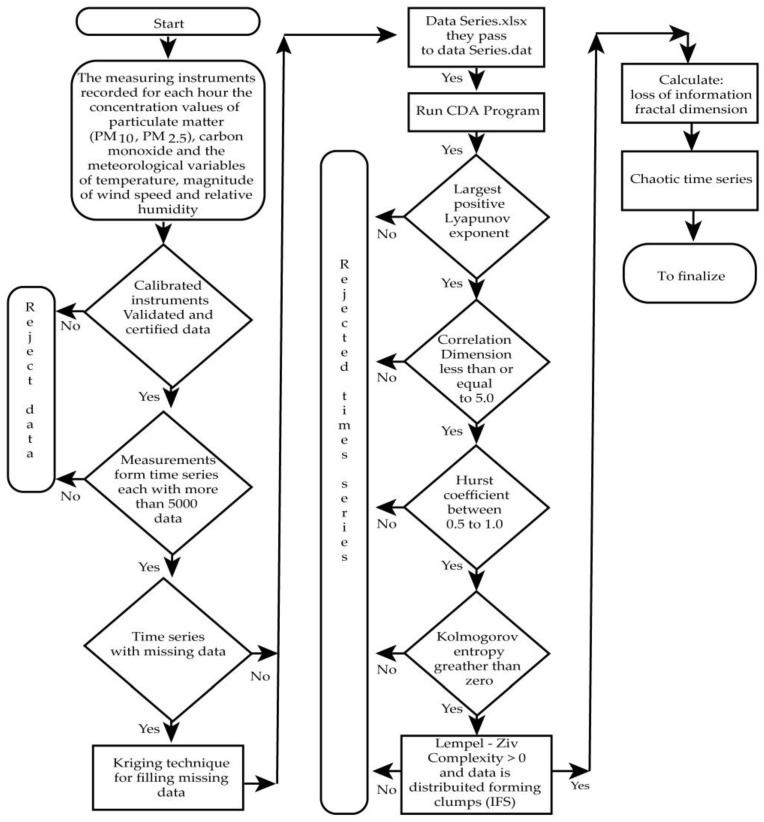
Flowchart that allows to verify if each time series is chaotic and calculates the chaotic parameters of interest (Lyapunov exponent, correlation dimension, entropy, Hurst exponent, Lempel-Ziv complexity, fractal dimension and loss of information).

**Figure 5 ijerph-20-03941-f005:**
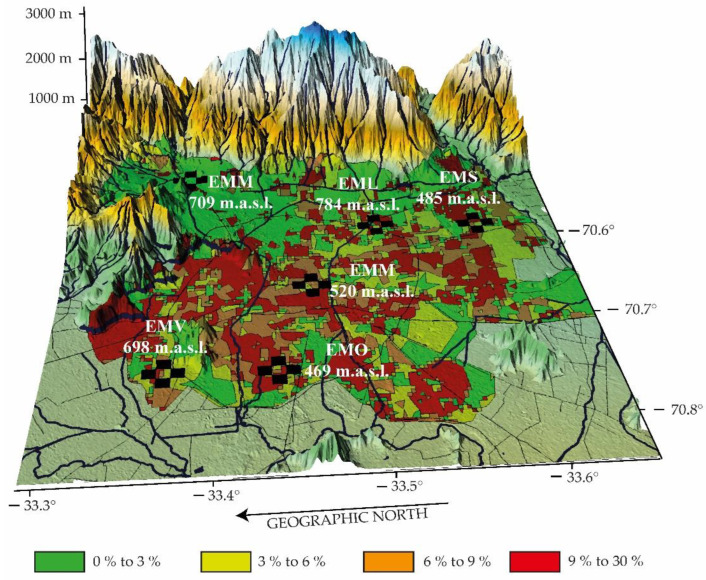
Geographical distribution of the monitoring stations for this study (network of black crosses) and high urban densification [37] (in color for the areas with the highest population density).

**Figure 7 ijerph-20-03941-f007:**
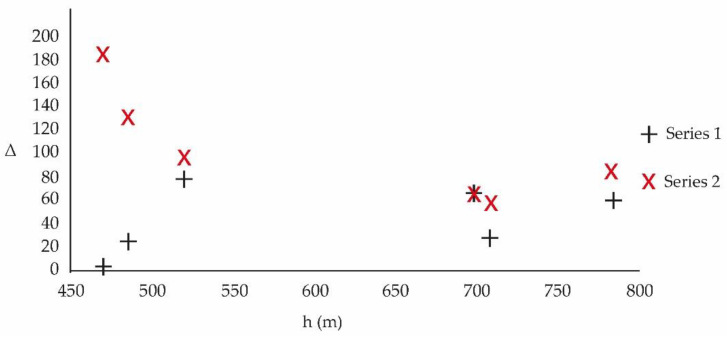
Represents the height of the different measurement locations and the difference in temporal heat between pollutants and meteorological variables according to periods 2010–2013 (Series 1) and 2017–2020 (Series 2) for a basin configuration.

**Figure 8 ijerph-20-03941-f008:**
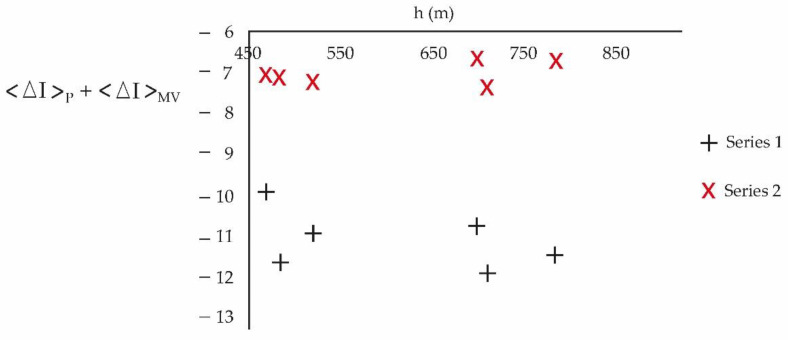
Series 1 corresponds to the loss of information for the period 2010–2013 and Series 2 corresponds to the loss of information for the period 2017–2020. This last period loses information faster (it is more chaotic).

**Table 1 ijerph-20-03941-t001:** The continuous recording of the variables in this study is carried out at 2 [m] above the ground in all sensors, with the exception of the one measuring the magnitude of wind speed, which is at 10 [m] [37].

Station Name	Location	PM10	PM2.5	CO	T	RH	WV	OWNER
1.La Florida, EML, masl:784 [m]	33°30′59.7″ S 70°35′17.4″ W	Attenuation Beta-Met One 1020	Attenuation Beta-Met One 1020	Gas Correlation Filter IR Photometry -Thermo 48i	VAISALA HMP35A	VAISALA HMP35A	Sensor-Met One 010C	SINCA
2.Las Condes, EMM, masl:709 [m]	33°22′35.8″ S 70°31′23.6″ W	Attenuation Beta-Met One 1020	Attenuation Beta-Met One 1020	Gas Correlation Filter IR Photometry -Thermo 48i	VAISALA HMP35A	VAISALA HMP35A	Sensor-Met One 010C	SINCA
3.Santiago- Parque O’Higgins, EMN, masl: 570 [m]	33°27′50.5″ S 70°39′38.5″ W	Attenuation Beta-Met One 1020	Attenuation Beta-Met One 1020	Gas Correlation Filter IR Photometry -Thermo 48i	VAISALA HMP35A	VAISALA HMP35A	Sensor-Met One 010C	SINCA
4.Pudahuel, EMO, masl:469 [m]	33°27′06.2″ S 70°40′07.8″ W	Attenuation Beta-Met One 1020	Attenuation Beta-Met One 1020	Gas Correlation Filter IR Photometry -Thermo 48i	VAISALA HMP35A	VAISALA HMP35A	Sensor-Met One 010C	SINCA
5.Puente Alto, EMS, masl:698 [m]	33°33′01.3″ S 70°34′51.4″ W	Attenuation Beta-Met One 1020	Attenuation Beta-Met One 1020	Gas Correlation Filter IR Photometry -Thermo 48i	VAISALA HMP35A	VAISALA HMP35A	Sensor-Met One 010C	SINCA
6.Quilicura, EMV, masl:485 [m]	33°21′51.6″ S 70°44′53.9″ W	Oscillating Element Microbalance TEOM-Thermo 1400AB	Attenuation Beta-Met One 1020	Gas Correlation Filter IR Photometry -Thermo 48i	VAISALA HMP35A	VAISALA HMP35A	Sensor-Met One 010C	SINCA

**Table 2 ijerph-20-03941-t002:** Magnitude of the average wind speed $\overset{-}{v}$ (m/s) according to the study periods (2010–2013 and 2017–2020) and location.

Station	PA (EMS)	LF (EML)	PO (EMN)	P (EMO)	Q (EMV)	LC (EMM)
2010–2013	1.7177	1.2042	1.2844	1.6740	1.6314	1.2682
2017–2020	1.4159	0.8896	1.0166	1.3191	1.2586	1.2054

**Table 3 ijerph-20-03941-t003:** For the period 2010–2013, in all the measurement locations, the average of ${\overset{-}{\gamma }}_{C}$, the addition and the height with respect to sea level are presented.

2010–2013:						
Station	PA (EMS)	LF (EML)	PO (EMN)	P (EMO)	Q (EMV)	LC (EMM)
Average (°C/h)	0.4192922	0.36230133	0.36009624	0.60208629	0.49100476	0.38847965
Addition (°C/h)	11,933.8947	10,311.8205	10,249.0592	17,136.58	13,974.9774	11,056.9079
Average (K/h)	273.6	273.5	273.5	273.8	273.6	273.54
Height (masl)	485	784	520	469	698	709

**Table 4 ijerph-20-03941-t004:** For the period 2017–2020, in all the measurement locations, the average of ${\overset{-}{\gamma }}_{C}$, the addition and the height with respect to sea level are presented.

2017–2020:						
Station	PA (EMS)	LF (EML)	PO (EMN)	P (EMO)	Q (EMV)	LC (EMM)
Average (°C/h)	0.92816795	0.61691359	0.65587701	0.95674275	0.94234843	0.79836877
Addition (°C/h)	26,418.4443	17,559.2116	18,668.2274	27,231.7689	26,822.0633	22,723.9704
Average (K/h)	274.1	273.80	273.81	274.1	274.1	274.0
Height (masl)	485	784	520	469	698	709

**Table 5 ijerph-20-03941-t005:** Presents a summary of ${\overset{-}{\gamma }}_{C}$ according to locality (commune), height (referred to sea level), average temperature and the increase in temperature per period. The values in columns 5 and 6 were approximated.

		$\mathrm{Series}\;1\;{\overset{-}{\gamma }}_{C}$ (K/h)	$\mathrm{Series}\;2\;{\overset{-}{\gamma }}_{C}$ (K/h)	$\overset{-}{T}$ + Average (°C)	$\overset{-}{T}$ + Average (°C)
Station	Height (masl)	2010–2013	2017–2020	2010–2013	2017–2020
EML	784	273.50	273.80	15.40 + 0.36 = 15.8	16.12 + 0.93 = 17.1
EMM *	709	273.54	274.00	15.86 + 0.40 = 16.3	15.57 + 0.62 = 16.2
EMV	698	273.60	274.10	15.77 + 0.50 = 16.3	16.85 + 0.66 = 17.5
EMN	520	273.50	273.81	15.34 + 0.36 = 15.7	16.17 + 0.96 = 17.1
EMS	485	273.60	274.10	14.69 + 0.42 = 15.1	15.53 + 0.94 = 16.5
EMO	469	273.80	274.10	16.77 + 0.60 = 17.4	16.80 + 0.80 = 17.6

* corresponds to the commune of Las Condes (belongs to the small group of communes with the highest income in Chile), which has made mitigation investments (its communal regulatory plan imposes restrictions on the construction of high-rise buildings).

**Table 6 ijerph-20-03941-t006:** Contains the average temperature for the measurement period 2010–2013 and the difference between temporal heat variation between pollutants and meteorological variables using Equation (30). The last row are average values.

h (masl)	$\overset{-}{T}$ (°C)	S_KP_ (1/h)	S_KMV_ (1/h)	(δQ/dt)_P_ (K/h)	(δQ/dt)_MV_ (K/h)	Δ(K/h)
784 (EML)	15.40	1.542	1.334	444.9441	384.9257	60.02
709 (EMM)	15.86	1.550	1.452	447.9655	419.6425	28.00
698 (EMV)	15.77	1.431	1.202	413.4445	347.2818	66.10
520 (EMN)	15.34	1.531	1.262	441.6782	364.0744	78.00
485 (EMS)	14.69	1.377	1.289	396.3557	371.0258	25.33
469 (EMO)	16.77	1.210	1.197	350.8032	347.0342	3.80
	15.64			415.8652	372.3307	43.54

**Table 7 ijerph-20-03941-t007:** Contains the average temperature for the measurement period 2017–2020 and the difference between temporary heat variation between pollutants and meteorological variables using Equation (30). The last row are average values.

h (masl)	$\overset{-}{T}$ (°C)	S_KC_ (1/h)	S_KM_ (1/h)	(δQ/dt)_C_ (K/h)	(δQ/dt)_M_ (K/h)	Δ(K/h)
784 (EML)	16.12	1.577	1.284	456.1788	371.4226	84.7561
709 (EMM)	15.57	1.406	1.205	405.9403	347.9076	58.0327
698 (EMV)	16.85	1.220	0.994	353.8000	288.2600	65.5400
520 (EMN)	16.17	1.479	1.145	427.9043	331.2714	96.6329
485 (EMS)	15.53	1.702	1.250	491.3333	360.8500	130.4833
469 (EMO)	16.80	1.630	0.993	472.6185	287.9204	184.6982
	16.20			434.6292	331.2720	103.3572

**Table 8 ijerph-20-03941-t008:** Height according to measurement locations and heat difference between pollutants and urban meteorology.

	2010–2013	2017–2020
h (m)	Δ(Series 1)	Δ(Series 2)
784	60.02	84.7561
709	28.00	58.0327
698	66.10	65.5400
520	78.00	96.6329
485	25.33	130.4833
469	3.80	184.6982

**Table 9 ijerph-20-03941-t009:** Total loss of information adding the loss of information of the pollutants and the loss of information of the meteorological variables in the two periods of the study.

		Series 1	Series 2
Station	h (m)	2010–2013	2017–2020
EML	784	−11.42	−6.694
EMM	709	−11.898	−7.312
EMV	698	−10.724	−6.647
EMN	520	−10.894	−7.234
EMS	485	−11.611	−7.076
EMO	469	−9.904	−7.063

## Data Availability

The data were obtained from the public network for online monitoring of air pollutant concentration and meteorological variables. The network is distributed throughout all of Chile, without access restrictions. It is the responsibility of SINCA, the National Air Quality Information System, dependent on the Environment Ministry of Chile. The data for the two study periods will be available for free use on the WEB page: URL: https://sinca.mma.gob.cl accessed on 4 April 2021.

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
