# Peer review of "Intensive Urbanization, Urban Meteorology and Air Pollutants: Effects on the Temperature of a City in a Basin Geography"

_ijerph, 2023, doi:10.3390/ijerph20053941_

Round 1

Reviewer 1 Report

I have many suggestions to point out in order to improve this manuscript. All detailed in the pdf document that is attached below. Almost all suggestions are minor corrections, but, for improving significatively the manuscript, all ítem pointed out should be carefully considered/corrected.

Reviewer 2 Report

Dear Authors,

I appreciate your work, although the research topic is not new. I thought the research method presented was appropriate. The layout of the work is clear and correct. Figures and tables with results are clear. The results contained in the paper (after additions and corrections) can be a valuable aid and reference for other metropolitan areas.

Points to be clarified/improved:

1) I suggest shortening the title of the manuscript.

2) What determined the choice of research period: 2010 - 2013 and 2017 - 2020? And what happened in the period 2014 - 2016?

3) L38-L80

1.2. Heat and human health

Authors describe the negative impact of air quality in a city with many millions of inhabitants and cite many research papers. I consider that Authors could have added what is the percentage increase in the various observed ailments and illnesses of the inhabitants.

4) I believe that the paper lacked an indication of the practical aspects: What are the most urgent needs and opportunities to improve air quality conditions in Santiago de Chile?

5) L94:

The number of subsection 2.1.1. should be preceded by the number of subsection 2.1

6) The title should begin with a capital letter.

L94: „thermal conduction” --> „Thermal conduction”

L 262:

4.1. discretization --> 4.1. Discretization

L288:

4.2. irreversible processes --> 4.2. Irreversible processes

7) Tables 2, 3, 4 should be preceded by a mention in the text of the manuscript, as has been done for Table 1.

8) L 435: Patents

Please complete the patent information or remove the chapter title.

Reviewer 3 Report

This is a well drafted paper.

For Introduction, I'd probably start with a little bit more context on the importance of the problem rather than starting directly with definitions.

I think the theoretical basis section has its place, but after a long and complex series of mathematical equations, it is important to reorient the reader with the actual research question we are trying to answer.

We need to specifically know why these data points were chosen in the methods.

The Results are not structured well. We need a section/table/diagram on how the two subsections tie together.

Discussion should be 5 instead of 4.

Conclusions should be 6 instead of 5.

Reviewer 4 Report

Submitted to the journal Int. J. Environ. Res. Public Health article was entitled: " Intensive urbanization, temperature increase and effects on the population. Study based on measurements of urban meteorology and air pollutants”.  In my opinion, the title of the article does not reflect its content.

The goal written in the paper is as follows: “The objective is to answer the question: is it possible to obtain indicators derived from both techniques, applied to time series, that analyze thermal transfer and that their predictions are similar, showing an increase in temperature and heat ? With this purpose, the fundamentals of both techniques will be briefly developed.” In fact, the entire work is devoted to the realization of such a goal, which has little relation to the title of the article.

The Authors studied two time series of meteorological data and concentrations of selected pollutants, however, in my opinion, these studies may bring new elements to the development of the theory of computational techniques, but they do not provide results and conclusions related to the issues contained in the title of the work.

The study did not use data on the health of the population, as well as indicators characterizing increasing urbanization. Meteorological data are not sufficient to assess temperature changes on their basis - two periods of only about 3 years were compared. The work lacks the characteristics of the climatological conditions of the study area (values from 30 years), therefore it is impossible to assess whether the first period was representative for many years, or perhaps - exceptionally cold?

No information about the meteorological data itself - were the "raw" data used or were they processed in some way? Lack of information about the measuring stations - are the stations of the same rank, do they use the same measuring devices, also were the measurements performed in the same way in both compared periods, were there data gaps, if so, how were they supplemented? What is the representativeness of the measuring stations?

 To what extent do differences (increases) in temperature reflect the impact of urbanization, and to what extent do global temperature rises reflect it?

Why was precipitation omitted in the research - very important for the concentration of atmospheric pollutants?

There is also no information on the method of measuring selected air pollutants - what kind of equipment was used, was it the same at all stations in the periods under study, were the measurements performed at the same stations as the meteorological measurements, what is the station's surroundings like?

In my opinion, the existence of so many doubts does not allow for a positive opinion of the work. I do not recommend this article for publication in the journal.

Detailed notes:

In chapter 1.2. Human heat and health no bioclimatology studies included, apart from Belkin research (lines 63-66), there should be a reference to indicators such as the heat stress index (HSI), air enthalpy, heat sensations based on the HUMIDEX index, the heat load index (HL).

Theoretical foundations (chapter 2) dominated the work, too little attention was paid by the Authors to describing the materials and methods as well as the results of the work.

Line 249 - I don't understand - are the measurements in the shade or in the sun?

Line 253 - lack of information on the average level of pollution in the research area, their main types, sources and the quantitative trend of pollution, as well as population density

Lines 255-256: please provide specific data on the increase in the number of vehicles.

Results: the tables are insufficiently described, there are no explanations for some of the abbreviations in the column headings (Tables 1-3, 5-7), Tab. 2-3: in the 3rd line (Average (K/h), the unit is repeated next to the data (it should be removed).

The tables description is largely a description of what the table contains, similar to table headers. There is no in-depth analysis of the data contained in tables.

Discussion:  Lines 387-391 - there is no information about waste in the results of the work, I do not know why reference was made to the literature on this subject [47].

Lines 395-400: have no reference to the results of the research conducted in this work. The Authors did not study the impact of meteorological conditions and pollution on health.

Conclusions: Lines 429-434 - this study did not document the harmful effects of rising temperature on human health, especially in psychological, emotional and biological aspects. This has been documented by other authors in their works cited here, but it is not an achievement of the Authors of this paper.

Round 2

Reviewer 4 Report

In my opinion, the Authors referred to the most important comments and introduced the necessary corrections. I accept the paper in its current form and recommend its publication in the journal.